# The Forgotten Agenda of Wasting in Southeast Asia: Burden, Determinants and Overlap with Stunting: A Review of Nationally Representative Cross-Sectional Demographic and Health Surveys in Six Countries

**DOI:** 10.3390/nu12020559

**Published:** 2020-02-20

**Authors:** Mueni Mutunga, Severine Frison, Matteo Rava, Paluku Bahwere

**Affiliations:** 1United Nations Children’s Fund (UNICEF) East Asia Pacific Regional Office, Bangkok 10200, Thailand; 2Department of Infectious Disease Epidemiology, London School of Hygiene and Tropical Medicine (LSHTM), London WC1E 7HT, UK; severine.frison@lshtm.ac.uk; 3Bergen Center for Ethics and Priority Setting (BCEPS), University of Bergen, 5009 Bergen, Norway; matteorava.moh@gmail.com; 4Centre de Recherche en Epidémiologie, Biostatistique et Recherche Clinique, Ecole de santé publique, Université Libre de Bruxelles, 1080 Brussels, Belgium; paluku.bahwere@ulb.ac.be

**Keywords:** wasting, severe wasting, wasting and stunting, prevalence, burden, Southeast Asia, DHS, MICS, risk factors, under five

## Abstract

Childhood wasting is among the most prevalent forms of undernutrition globally. The Southeast Asia region is home to many wasted children, but wasting is not recognized as a public health problem and its epidemiology is yet to be fully examined. This analysis aimed to determine the burden of wasting, its predictors, and the level of wasting and stunting concurrence. Datasets from Demographic and Health Surveys and Multiple Indicator Cluster Surveys in six countries in the region were analyzed. The pooled weighted prevalence for wasting and concurrent wasting and stunting among children 0–59 months in the six countries was 8.9%, 95% CI (8.0–9.9) and 1.6%, 95% CI (1.5–1.8), respectively. This prevalence is approximately 12-fold higher than the 0.7% prevalence of high-income countries; and translated into an absolute number of 1,088,747 children affected by wasting and 272,563 concurrent wasting and stunting. Wasting prevalence was 50 percent higher in the 0–23-month age group. Predictors for wasting included source of drinking water, wealth index, urban residence, child’s age and history of illness and mother’s body mass index. In conclusion, our analysis showed that wasting is a serious public health problem in the region that should be addressed urgently using both preventive and curative approaches.

## 1. Introduction

Undernutrition in the first 1000 days post-conception has both short- and long-term detrimental consequences for the health and nutrition status of children, and adversely affects the economic productivity of nations [1]. It is associated with cognitive deficits that lead to lower educational performance [1,2,3,4,5], physical growth deficits that can limit economic productivity in adulthood [1,6,7], immune system dysfunction and reduced efficacy of vaccines—hence increased susceptibility to and severity of infections [1,8,9]. Among women, the consequences of undernutrition during childhood can persist until reproductive age and cause intrauterine growth restriction [1,10]. Wasting, a form of acute malnutrition, is diagnosed in children 6 to 59 months old when the weight-for-height z-score (WHZ) of WHO child growth standards is <−2 standard deviations (SD). It is one of the most prevalent forms of undernutrition, is associated with high short-term mortality and some studies have shown that it also contributes significantly to linear growth restriction [11,12,13]. Studies have shown that a child with severe wasting, also called severe acute malnutrition (SAM), is up to 12 times more likely to die than a well-nourished child [14,15,16], and the survivors of SAM episodes have an increased risk of developing non-communicable chronic diseases during adulthood [1,17,18,19,20]. Recent studies have shown that children who are both wasted and stunted are at a higher risk of death than those with only one of the two nutritional deficits [21].

The standardization of the SAM management, by the World Health Organization (WHO) and the adoption of the community-based management of acute malnutrition (CMAM) have contributed to the tremendous reduction in SAM case fatality [22,23,24,25,26]. However, the coverage of interventions addressing SAM remains low in the Southeast Asia region as observed by Ahmed et al. in their review paper published in 2014 that covered, Cambodia, Myanmar, Timor-Leste, Lao PDR and Vietnam among other countries in the Asia region [27]. This is despite the region being widely known to be home to a large number of wasted children [28]. Studies have demonstrated that wasted Asian children, regardless of the region, respond well to treatment based on the WHO protocol and that premature interruption of this treatment is associated with increased risk of death during the weeks following the interruption [27,29,30,31,32,33]. The low coverage of SAM curative interventions is partly due to the belief that SAM observed in Asian children has different determinants and clinical features than those observed in African children [34,35,36,37,38]. This position shared by many policymakers in the Southeast Asia region limits provision of services to address wasting [5,27].

This article is based on a review of nationally representative surveys of selected countries in the Southeast Asia region to determine wasting burden, determinants and the overlap with stunting. It aims to contribute to the existing and growing body of evidence needed to support the prioritization of treatment of wasting in addition to the ongoing effort to prevent both wasting and stunting.

## 2. Materials and Methods

### 2.1. Study Design

The present study is a secondary data analysis of the latest Demographic and Health Survey (DHS) and Multiple Indicator Cluster Surveys (MICS) or National Food and Nutrition Survey (NFNS) (when DHS or MICS data is unavailable) from Cambodia, Lao PDR, Myanmar, Thailand, Timor-Leste and Vietnam.

### 2.2. Datasets

Data were from nationally representative cross-sectional surveys conducted between 2013 and 2017 in six Southeast Asian countries that had publicly available data. All the countries in the region were eligible, but nationally representative survey data were not publicly available all countries. These datasets used included the Cambodia Demographic and Health Survey (DHS) 2014, Lao PDR Multiple Indicator Cluster Surveys (MICS) 2017, Myanmar DHS 2015, Thailand MICS 2015/6, Timor-Leste National Food and Nutrition Survey (NFNS) 2013 and Vietnam MICS 2011.

All the surveys used multistage cluster sampling. Detailed sampling plans are available from the final survey reports. All the datasets included sampling weights used for the calculation of nationally representative statistics. For this regional analysis, data from each country were imported into Stata14.1 (Stata Corp) and merged for analysis.

### 2.3. Analytic Sample

Children under five years of age were included in the analysis if they had a plausible value for WHZ (i.e., ranging from −5 to 5).

### 2.4. Data Management and Analysis

The primary outcomes for our analyses were wasting, stunting, and concurrence of wasting and stunting. Wasting was diagnosed when WHZ of WHO child growth standards was <−2 SD, and severe wasting when WHZ was <−3 SD and <−2. Stunting was diagnosed when length/height-for-age z-score (HAZ) was <−2 SD and severe stunting when HAZ was <−3 SD. Wasting and stunting concurrence was diagnosed when both WHZ and HAZ are <−2 SD. Entries with missing values of the child age, sex and any anthropometric measurement were excluded in the analysis. A two-stage approach to anthropometric data cleaning was applied. First we applied the biological plausibility criteria where values were set to missing if weight > 50 kg or if height > 200 cm. Second, we applied the WHO statistical probability criteria where HAZ was set to missing if HAZ > 6 or <−6 and WHZ was set to missing if WHZ > 5 or <−5. Additionally, any record with missing parameters for the calculation of WHZ or HAZ was dropped from the analysis.

### 2.5. New Variable Creation 

Most variables were used as stored in the survey datasets and are presented in Table 1. Variables transformed to create a new variable were: child’s age, for which six categories were created as per DHS and MICS categorization (Table 2); immunization status, for which a category “unsure” was added to minimize missing records for children for whom it was reported that they had a vaccination card, but the immunization timeliness could not be ascertained; and, number of antenatal care visits, for which a category “other” was added for respondents whose antenatal care visits were reported, but they could not fit into the category of being attended by a skilled staff or not.

### 2.6. Data Analysis

We calculated pooled and individual countries weighted prevalence of wasting, severe wasting and concurrence of wasting and stunting. We also calculated the burden (number of children affected) overall and for each country. To calculate the burden of wasting and concurrent wasting and stunting, we used the 0 to 59-month-old population for the corresponding country and survey year from the State of the World Children annual reports. The burden of wasting, severe wasting and concurrence of wasting and stunting were then estimated by multiplying the observed prevalence by the total population estimate for 0 to 59-month-old children.

We used univariate and multivariate multilevel logistic regression to identify individual, household, maternal, and child characteristics significantly associated with wasting in the region. First, we developed three separate models for each level of characteristics, namely household model, mother model and child model. Subsequently, we constructed a combined model assessing all the covariates together. For all the models, we included all the variables available for analysis into the dataset and excluded only the covariates introducing collinearity. For all the combined models, we also excluded covariates associated with higher than 10% sample size reduction. No stepwise method was used, and significant and non-significant covariates are presented in the tables to show the effect of adjustment for all the covariates. 

### 2.7. Ethical Considerations

Ethical approval for the analyses presented in this paper was not sought as the paper is based on data obtained after completed the mandatory registration to DHS Macro (http://dhsprogram.com/data/Access-Instructions.cfm) and from United Nations Children’s Fund (UNICEF) for Multi-indicator Cluster Surveys that serve as authorization to access the datasets for secondary analysis research. The authorization to use the Timor-Leste NFNS data was obtained from the National Office of Statistics, Timor-Leste.

## 3. Results

### 3.1. Characteristics of Survey Households, Mothers, and Children

Table 1 describes the surveys analyzed for this paper. The surveys were conducted between 2011 and 2017 with the Vietnam survey being the oldest and the Thailand survey the most recent.

The number of households and mothers of children under five years old interviewed varied country by country with Vietnam having the smallest sample and Thailand the largest. Except for Thailand and Vietnam, two-thirds of the children included in the analysis were from rural areas. The 20 to 34 years age group dominated the sample of mothers interviewed in all the countries and represented >60% of those interviewed in most of the countries. Overall, adolescent mothers represented less than 5% of the interviewed mothers. There was great variation in the level of maternal formal education. In Thailand and Vietnam most mothers attained secondary school or higher, while in the other countries over 60% had either primary or no formal education. Of the three countries for which data were available, Timor-Leste had the highest percentage of mothers with short stature and BMI < 18.5 kg.m^−2^. The sex ratio of surveyed children and their distribution across the different age groups was balanced in almost all the countries surveyed with each year interval contributing around 20% of the sample of children included in the analysis.

### 3.2. Prevalence and Burden of Wasting

Table 2 shows the pooled and country prevalence of wasting and severe wasting. The pooled prevalence of wasting was more than 5%. For individual countries, this prevalence was above 5% in five of the six countries examined. Timor-Leste and Cambodia had a high prevalence while Lao PDR, Myanmar and Thailand had a medium prevalence. For severe wasting prevalence, the pooled prevalence of the six countries reached the emergency threshold of 2%. The prevalence of wasting was above this threshold in Cambodia, Lao PDR and Timor-Leste and lower than this threshold in Myanmar, Thailand and Vietnam. In all the six countries, the number of children affected was high, giving a pooled figure of over 1 million under-five children affected by wasting, with close to 280,000 of them being severely wasted.

The prevalence of wasting across the age groups did not follow the same pattern across the countries included in the analysis (Figure 1). Cambodia, Myanmar, Thailand and Vietnam had the highest prevalence of wasting and severe wasting in the 0 to 5 months age group. In Lao PDR and Timor-Leste, the peak prevalence of wasting was highest among the 6–11 month and 12–23 month age groups respectively while for severe wasting the peak was in the 6–11 months group. In Cambodia, Lao PDR, Myanmar and Timor-Leste, multiple age groups had a prevalence of severe wasting above 2% with all six-age groups in Lao PDR having a prevalence above this cut-off. For Vietnam, no age group had a prevalence of severe wasting above 2%. 

When the age was dichotomized to <24 months and ≥4 months, the pooled prevalence (95% CI) of wasting for the 0 to 23 months age group and for the 24 to 59 months age group were 12.0 (10.8–13.4%) and 6.3 (5.5–7.2%), respectively (*p* < 0.001). For severe wasting, the prevalence was 2.9 (2.3–3.6%) for the 0 to 23 months age group and 1.3 (0.9–1.8%) for the 24 to 59 months age group (*p* < 0.001). 

The prevalence of wasting and severe wasting observed in these age groups by country are presented in supplementary Appendix A. The difference between the two age groups was highly statistically significant (*p* < 0.001) in all the countries for both wasting and severe wasting. The relative risk of a child being wasted in the 0 to 23 months age group in comparison to a child in the 0 to 59 months varied across countries with the highest observed relative risk in Timor-Leste (RR 95% CI = 2.13 (1.88–2.41%)) and the lowest in Thailand (RR 95% CI = 1.24 (1.05–1.45%)). For severe wasting, Timor-Leste again had the highest relative risk (RR 95% CI = 3.23 (2.35–4.45%)) while Vietnam had the lowest (RR 95% CI = 1.15 (0.63–2.10%).

The share of the total number of wasted children was equal for children in 0 to 23 months and 24 to 59 months age groups in the pooled analysis (Appendix A). The country by country analysis showed that this was also the case for Cambodia, Myanmar and Vietnam. However, in Lao PDR and Thailand, the 0 to 23 months age group contributed more cases while the 24 to 59 months age contributed more cases in Timor-Leste (Appendix A).

### 3.3. Prevalence and Burden of the Concurrence of Wasting and Stunting 

The prevalence of concurrence of wasting and stunting varied widely from 1.05% to 5.30% and was the highest in Timor-Leste and lowest in Vietnam (Table 3). For the six countries, over 250,000 children were both wasted and stunted at the time of the surveys with the figure varying across countries and ranging from 10,070 to 75,443 children wasted and stunted (Table 3). Close to 18,000 children in the 0 to 59 months age group were experiencing both severe wasting and severe stunting at the time of the surveys with Thailand having the highest number and Lao PDR the lowest number.

There was a perfect and direct correlation (Spearman rho correlation rs = 1.0; *p* < 0.001) between the country’s prevalence of wasting and concurrence of wasting and stunting (Figure 2). 

The concurrence was the lowest in the 0 to 5 month age group in four of the six countries (Figure 3). This prevalence shows a significant linear trend increasing as age increased in Cambodia (*p* = 0.001), Lao PDR (*p* = 0.035), in Myanmar (*p* = 0.016), Timor-Leste (*p* = 0.001) and Vietnam (*p* = 0.033) but not in Thailand (*p* = 0.457). Despite the statistically significant linear trend, the highest prevalence was not always in the 48 to 59 age group. The peak prevalence for concurrence was observed in 36 to 47 months age group in Cambodia, in 12 to 23 months age group in Lao PDR, Thailand and Timor-Leste and in 48–59 months age group in Myanmar and Vietnam. When we excluded children below 12 months of age in the analysis, the direct linear relationship for the 12–59 age group was observed only for Timor-Leste (*p* < 0.001). The test of the linear trend was not significant for Cambodia (*p* = 0.920), Lao PDR (*p* = 0.115), Myanmar (*p* = 0.993), Thailand (*p* = 0.571) and Vietnam (*p* = 0.168).

The burden of concurrence was equally distributed across the under-five age groups for Cambodia, Lao PDR, Myanmar and Vietnam with each 12-month interval encountering approximately 20% share of the burden (Appendix A). For Thailand, the age group 0 to 11 months had a much lower share of concurrent wasting and stunting than the other four age groups and for Timor-Leste, the 0 to 11 month and 12 to 23 month age groups contributed 50% of the share of the burden while the 48 to 59 age group contributed only around 10% (Appendix A).

### 3.4. Factors Associated with Risk of Wasting 

Table 4 presents the associations between wasting and the potential predictors tested in analyses that considered households, maternal and child characteristics separately. The household characteristic model showed that all the factors included in the analysis—place of residence, wealth index, number of people in the household source of drinking water were independently associated with risk of a child being wasted (Table 4). Comparing the richest and poorest wealth quintiles showed that children from the poorest quintile had a 25% increase in the risk of being wasted (OR (95%) = 1.25 (1.10–1.43%); *p* = 0.001). 

The mother’s characteristics model identified mother’s age and mother level of education as independent predictors of wasting among children (Table 4). Marital status was not an independent risk factor for wasting (Table 4). The mother characteristics model that included maternal BMI and height in addition to all the other variables revealed that the risk of being wasted for a child of a mother with a normal BMI (≥18 kg.m^−2^ and <25 kg m^−2^) differed significantly with that of a mother with low BMI or high BMI. The risk of being wasted was 69% higher for a child of a mother with low BMI < 18.5 kg m^−2^ (AOR (95%CI) = 1.65 (1.50–1.90%); *p* < 0.001) when compared to a child of a mother with normal BMI. A child of a mother with higher BMI than the reference had a lower risk of being wasted (AOR (95% CI) = 0.76 (0.63%–0.91)); *p* = 0.003) for a child of a mother with BMI ≥ 25 kg.m^−2^ and BMI < 30 kg m^−2^, and AOR (95% CI) = 0.55 (0.34–0.87%); *p* = 0.0.011 for a child of a mother with BMI ≥ 30 kg m^−2^. Maternal short stature, height 145 cm was not associated with a higher risk of wasting (AOR (95% CI) = 0.97 (0.83–1.15%); *p* = 0.767). In this model, including only data from Cambodia, Myanmar and Timor-Leste, mother’s level of education was no longer a predictor of child wasting, but mother’s age remained (data not shown).

The child characteristics model identified child’s sex, age, history of fever and or diarrhea as independent predictors of wasting (Table 4). Size at birth and quality of antenatal care were also included in the model for three countries and were independently associated with the presence of wasting. The risk of being wasted was significantly higher if the child was reported to have been very small at birth (AOR (95%) = 2.01 (1.24–3.27%); *p* = 0.004) or smaller (AOR (95%) = 1.56 (1.28–1.90%); *p* < 0.001) than average at birth. On the contrary, the risk of being wasted was reduced if the child was large (AOR (95%) = 0.69 (0.57–0.82%); *p* < 0.001) or very large (AOR (95%) = 0.54 (0.33–0.90); *p* = 0.017) compared to the average at birth according to mother’s recall. The variables of adjustment for this model were age, sex, vaccination status, antenatal care, history of fever and history of diarrhea. 

Interestingly, while children of the poorest households had an increased risk of being wasted compared to those from households of the average wealth category in the household characteristic model, there was no difference in the combined model between these two categories (Table 4). Another important change in association when comparing the combined and separate models is that in the mother characteristic model risk of wasting was lower among children with mothers over 20, compared to mothers who were less than 20, but this difference was not observed in the combined model.

In contrast, the differences between the average wealth category and the wealthiest categories that were not significant in the separate model became significant in the combined model with children in the wealthiest categories having an increased risk of wasting (Table 4). The combined model showed an association between higher levels of formal education and lower wasting in chidren, while child history of fever or diarrhea increased the likelihood of wasting even after controlling for the selected household’s and mother’s characteristics (Table 4).

## 4. Discussion

The analysis presented in this paper, based on nationally representative data of six countries of Southeast Asia, shows that the prevalence of wasting is of medium public health importance in the region. However, translating prevalence into an absolute number of children under five that are wasted highlights that wasting should be considered a serious public health problem for the region. The analysis also revealed that the prevalence of concurrent stunting and wasting is relatively low at 1.65%, but the overall number of children affected was more than 250,000 in the six countries alone. These findings call for increased attention to this health condition, given the associated higher risk of death. The analysis also showed that the age groups at which prevalence of wasting peaks varies across countries in the region, four countries showed a peak prevalence of wasting in children 0 to 5 months and two countries had the highest prevalence among children 6–23 months. Finally, the combined multivariate modelling of households, maternal and child variables identified antenatal care by skilled staff, maternal BMI and child size at birth as independent predictors of wasting, mediated through prenatal growth restriction, the source of drinking water, and history of recent episode of fever and diarrhea, that operate through postnatal growth deterioration and place of residence, wealth level of the household, mother’s formal education level and child’s sex.

### 4.1. Prevalence and Burden of Wasting

Our analysis has demonstrated that a significant proportion of under five children in the countries surveyed, are still suffering from wasting despite the rapid economic growth and the tremendous improvement of food availability [36,39]. Indeed, the pooled prevalence of 8.9% is 12-fold higher than the 0.7% for high-income countries [28]; and the prevalence is over 10% in two of the six countries examined. The total number of children affected by wasting is over one million, which is more than the total population of under-five children of Lao PDR and Timor-Leste combined. These findings justify classifying wasting as serious public health problem in the Southeast Asia region [28,40]. Furthermore, positioning wasting as a serious public health concern across the entire Southeast Asia region and tackling it is likely to yield global benefits as the region contributes a significant share to the global burden of wasting. Our analysis suggests that the six countries examined contribute approximately 2% of the global wasting burden, and it is estimated that the entire Southeast Asia region contributes to around 5.1 million cases or 10% of the global wasting burden [28]. This contribution, when added to that of South Asia, an equally densely populated region, in which wasting prevalence was between 9.5% and 21.0% in an analysis carried out recently by Harding et al., makes Asia the continent with the largest share of the global under five children wasting burden [41].

Our analysis has also shown that using prevalence figures for wasting solely to determine the public health importance of wasting may be misleading, especially in countries with a large population. For example, in Vietnam, using a wasting prevalence of 4.1%, wasting would be considered of low public health importance. However, the country has a higher absolute number of affected children than both Cambodia and Timor-Leste, which have wasting prevalence of 10.0% and 10.6%, respectively. Similar findings have been reported in seven countries of Southeast Asia, including four of the countries we analyzed, namely Cambodia, Lao PDR, Timor-Leste, Vietnam and Myanmar, Indonesia and the Philippines [42]. Consequently, the prevalence of wasting should always be contextualized. Many experts recommend using national or sub-national population size and national health system capacity to cope with the caseload, in addition to prevalence and aggravating factors for wasting in considering the classification and design of interventions to address childhood wasting. 

### 4.2. The Concurrence of Wasting and Stunting

Stunting is also a serious public health problem in Southeast Asia region, with most countries having a stunting prevalence of above 30% [42,43,44] and despite the coexistence of wasting and stunting in the region, no attention has been paid to the concurrence of both. Our analysis shows that the prevalence of concurrent wasting and stunting is low for the six countries examined, with a pooled prevalence of 1.65%. This prevalence was lower than the global prevalence of 3.0% reported by Khara et al. based on an analysis of 84 DHS survey including Cambodia, Lao PDR, Thailand and Timor-Leste [45]. It was also much lower than the pooled unweighted prevalence of 6.11% reported by Harding et al. for six countries in the neighboring South Asia region (Afghanistan, Bangladesh, India, Maldives, Nepal and Pakistan), confirming that the two Asian regions have different undernutrition profiles [41]. However, this unweighted prevalence for South Asia must be considered with caution as India provided 89% of the sample analyzed and alone had a concurrence prevalence of 6.62% [41]. 

As with the prevalence of wasting, interpretation of this low concurrence of wasting and stunting prevalence should take into account the population size of Southeast Asia. Indeed, the prevalence of 1.65% obtained in our analysis translates into a quarter-million of under five children affected. These children need particular attention as studies have shown that they are at high risk of dying even when they are moderately wasted [21,46]. Thus, the approach for screening and selecting children for wasting interventions should be adjusted to ensure children with concurrent wasting and stunting are able to access the most effective treatment for their condition. Some authors suggest to treat them as severely wasted children with the addition of weight-for-age screening criteria, where MUAC alone is used as eligibility for therapeutic feeding programs [21,47]. According to emerging evidence, the addition of a weight-for-age criterion allows the identification of concurrently wasted and stunted children and wasted children at increased risk of death that are not identified by current MUAC criteria [47].

Lastly, the observed low prevalence of concurrent stunting and wasting at less than 2% has important policy and programmatic implications for the region. It suggests that in Southeast Asia region, the great majority of wasted children will not be reached by programs designed to combat stunting alone and those that only use height/length-for-age criteria for the selection of beneficiaries. Both conditions should be targeted specifically, although programmatically interventions can be combined to maximize the impact. Indeed, although limited, there is evidence that stunted and wasted children respond well to treatment of wasting and that recovery from wasting may be followed by linear growth acceleration [12,48].

Our analysis shows that in most countries reviewed the prevalence of concurrence was lower among infants below 12 months of age, similar to the observation by Khara et al. who observed increased concurrence prevalence after 12 months of age [45]. These authors concluded that the increase in concurrence after 12 months was due to the high prevalence of both wasting and stunting in their study population. This explanation does not hold for our analysis as the increase in concurrence of wasting and stunting was also observed in countries where wasting prevalence was the highest amongst children less than 12 months of age. Alternative explanations are a rapid increase in the prevalence of stunting and an interrelationship between the two forms of undernutrition with stunting increasing risk of wasting occurrence and vice versa. The variation and degree of this interrelation may also explain the global and country level differences identified in wasting and stunting prevalence [21,45]. Unfortunately, the cross-sectional nature of the surveys included in our analysis do not allow for the verification of this hypothesis. This hypothesis should be verified in future appropriate longitudinal studies.

### 4.3. Risk Factors for Wasting

Our analysis has shown that the source of drinking water had the strongest association with wasting in the full model, including all the available variables and data from all the six countries. This finding is consistent with the literature, including studies conducted in a different countries in Asia [1,49,50]. The possible pathway is through increased frequency of diarrhea and intestinal parasites which contribute to the development of Environmental Enteric dysfunction syndrome [51,52,53,54]. Interestingly, diarrhea was also an independent predictor of wasting in this model. These findings advocate for the prioritization of programs aimed at improving access to clean water, despite the current weakness of evidence around prevention of undernutrition through WASH or combination of WASH and food supplementation interventions [50,55,56,57]. 

Several maternal factors have consistently been identified as factors associated with increased risk of wasting in under-five children. Younger maternal age, height below 145 cm, low level of education and low BMI has been linked with increased risk of wasting in several studies [35,41,58,59,60,61,62,63]. In our analysis, however, only BMI was associated with risk of wasting. There was no association between maternal age and maternal height and risk of child wasting. For the level of mother’s education, only those with more than secondary education had a lower risk of having a wasted child when compared to those without formal education. Low Maternal BMI is among the key determinants of low birth weight (LBW), also known as small size at birth. LBW prevalence is undoubtedly responsible for most of the observed association between low maternal BMI and wasting, although maternal BMI might have changed since the child was born [64,65]. The impact of LBW prevalence on wasting prevalence is evident in the four countries that had the highest wasting prevalence among infants 0 to 5 months old. This epidemiological profile is observed in many Asian countries [34,41]. Thus, strategies to tackle wasting in the Southeast Asia region must include prevention of LBW, including the prevention of malnutrition among all women of reproductive age (WRA) and adolescents. Targeting WRA with nutrition and health interventions could also break the cycle of undernutrition as women born LBW or who experience undernutrition during childhood have increased the chance of having a LBW infant or undernourished child [41,66,67,68]. Furthermore, preventing or treating undernutrition of WRA is likely to contribute to the reduction of the prevalence of non-communicable chronic diseases [69,70,71,72,73]. 

Size at birth was also an independent predictor of child wasting in a model including mothers’ BMI indicating that both variables had a direct and independent influence on the risk of wasting of under five children in the three countries that provided data for this analysis. A similar result has been reported by many other authors [41,74]. This suggest a difference in postnatal growth between LBW and normal birth weight infants. in [42,75,76]. Thus, efforts to reduce the prevalence of wasting among children under five should also include postnatal interventions aiming at improving the growth of LBW during early infancy. The World Health Organization (WHO) recommends exclusive breastfeeding up to six months of age and the provision of counselling to improve complementary feeding practices e [1,77]. Some nutrition experts are now advocating for complementing these mostly preventive interventions with community-based interventions that are able to reverse wasting among the children below six months of age [74,78]. Unfortunately, to date, there is no validated treatment approach for this during early infancy.

Our analysis showed persistent wasting prevalence of >5% among children 24–35 months in comparison to the reference group. This shows that that post-natal factors contribute to the occurrence of wasting in the six countries, and that these age groups also require policy attention. This view is also backed by the fact that 50% of the absolute number of wasted children were from the age group above 24 months. 

While wasting prevalence is below 5% in children 0–23 months, the increase in risk of wasting is greater than 50% for all age categories less than 24 months, when compared to the 48–59 month group. This finding underscores the higher vulnerability of children below 24 months of age to wasting. In line with the now widely accepted concept of 1000 day window of opportunity, children in the 0–23 months age group should be prioritized for all nutrition interventions [1].

A surprising finding from our analysis is the higher risk of wasting among children from urban households than those from rural households. Indeed, most studies have reported that rural children have a higher risk of becoming wasted than urban children [36,41]. Further investigations are needed to confirm and understand this relationship. However, the possibility of the shift towards more wasting in urban settings of low and middle-income countries was predicted in the UNICEF publication Innocenti Digest number 10 in 2002 [79]. Under five children living in informal settlements in large cities are often exposed to very precarious conditions that can lead to poor health status, including poor nutrition status [79,80]. The presence of such informal settlements should be a trigger for including urban setting in all strategies to tackle undernutrition.

The other variables associated with child wasting in our analysis are child sex, a recent episode of fever and a recent episode of diarrhea. For all these variables, our findings show a lower likelihood of being wasted for girls than boys and a higher likelihood of being wasted for those with a history of a recent episode of fever or diarrhea, consistent with the existing literature [1,41,42,81]. For fever and diarrhea, the mechanisms are well understood, and the interventions of known effectiveness are well integrated into the health system and implemented at scale in most countries. For the sex association with wasting, further studies are needed to understand this association. Some experts suggest that it could be an artefact due to the difference in new WHO growth standards and the median growth trajectories of boys. Other experts speculate that this could be due to a difference in physical activity.

The current global renewed interest in tacking undernutrition including wasting has not been sufficiently embraced by governments and other actors in the Southeast Asia region, yet it remains a serious public health problem [82,83]. Interventions to address wasting are unfortunately not currently accessible to many of the vulnerable children in the region and where offered the coverage is very limited [27]. Many Asian nutrition experts believe that prenatal factors, including poor maternal nutrition and health during the preconception period and pregnancy, are the main determinants of wasting in children under five. This is because the prevalence of wasting peaks in the 0–5 months age group in some Asian countries, compared to after infancy which is common in other settings [34,84]. The peak in prevalence of wasting in the 0 to 5 months group has recently emerged as a key indicator to decide whether the determinants of wasting in a given country are predominantly prenatal or postnatal [34,41,60]. In our analysis, the age distribution pattern across the different age groups was not consistent across all the countries. In Cambodia, Myanmar, Thailand and Vietnam wasting was highest in the 0–5 months age group as described in de Wagt et al. paper, but not in Lao PDR and Timor-Leste [34]. Moreover, in the four countries that had a higher prevalence of wasting in the 0 to 5 months group, and had a similar profile to that observed in South Asia, the prevalence of wasting remained above 5% in all the age groups up to the 48 to 59 months age group [34,41]. This can hardly be explained by the effect of prenatal factors alone [84]. Furthermore, the low overlap between stunting and wasting observed in this analysis also indicates that post-natal factors play an important role in the occurrence of wasting in the examined countries.

Lastly, the wasting prevalence of 8.9% we observed was encountered despite the rapid economic growth and food production increase experienced in the Southeast Asia region in recent years. This observation confirms that of several authors who previously reported that national-level indicators of economic situation and agriculture outputs can be misleading. An increase in macro-economic indicators such as the per capita gross domestic product may not have an immediate effect on nutrition indicators and the effect may remain limited years after the improvement of the income and food security national figures [39].

### 4.4. Strengths and Limitations

The results presented and discussed in this paper should be interpreted taking into account the strengths and limitations of our study. The main strengths of this study are the use of data from the most methodologically robust national surveys and the use of a multicounty dataset combining data of six countries in the Southeast Asia region. However, we cannot assume that these countries are representative of the entire region, given that they were selected according accessibility of data. Another strength is that we used the most recent surveys available at the time of the analysis.

Our study had several limitations. The first limitation arises from our study design (cross sectional data) that does not allow to distinguish causality from association. Neither is it possible to distinguish direction of association. Second, we acknowledge that our datasets were not designed for looking at different age groups and data is only representative of the under-five age group. All analysis by age group should be looked at with caution. Third, is the unavailability of data on the head of the household characteristics, household food security level, sanitation type, mothers’ principal source of income, child exclusive breastfeeding history and the child dietary diversity score. The high proportion of missing data for some key variables such as size at birth, mother’s height and BMI and child’s history of cough is also an important limitation. Four, is the variation in years of surveys. Data of the oldest survey was collected in 2011 and the most recent in 2017. Despite these limitations, we believe that the findings are informative and should serve as a basis for updating strategies to combat wasting in examined countries and all countries of the region facing this serious public health problem.

## 5. Conclusions

Wasting remains a serious public health problem in many countries of the Southeast Asia region. The global target to reduce the prevalence of wasting to below five per cent by 2025 may not be reached unless the policy environment is significantly improved to enable the scale-up of preventive and curative interventions with high impact on wasting incidence and nutrition outcomes. For a rapid decline in the prevalence of wasting, women of reproductive age (WRA) including adolescents, and children under five need to be targeted simultaneously with preventive and treatment approaches as both prenatal and postnatal factors are contributing to the burden of wasting in the region. Such integrated programming is likely to have an impact that goes beyond the control of wasting by reducing stunting and by contributing to the prevention of the looming non-communicable diseases pandemic. Our findings suggest that deliberate efforts must be made to address wasting despite the rapid economic growth and improvement in food security. Our recommendation is for countries in the region and their partners to immediately scale-up# interventions with proven efficacy to address wasting in children. Further studies should be undertaken to assess the causal pathways of wasting and propose the most appropriate strategies that support the elimination of wasting and other forms of malnutrition in the region.

## Figures and Tables

**Figure 1 nutrients-12-00559-f001:**
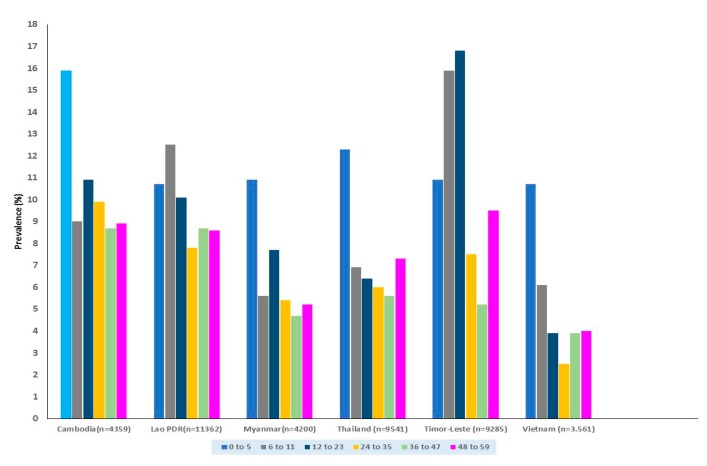
Prevalence of wasting in the different age groups by country.

**Figure 2 nutrients-12-00559-f002:**
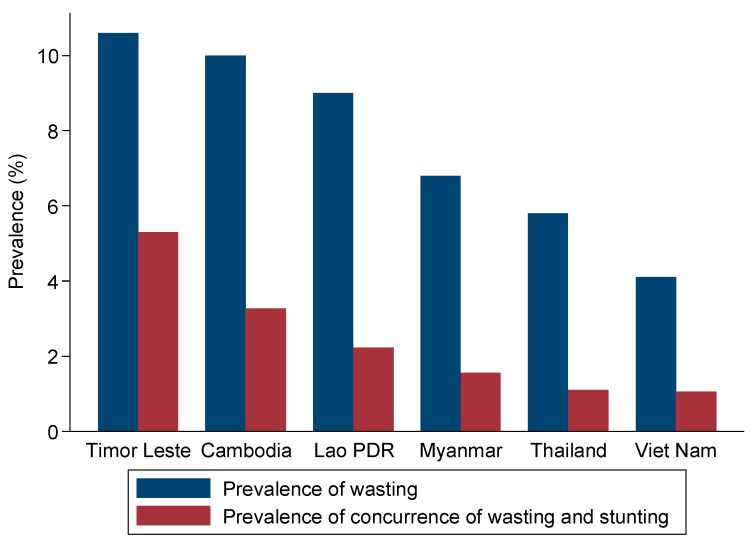
Prevalence of wasting and concurrence of wasting and stunting by country.

**Figure 3 nutrients-12-00559-f003:**
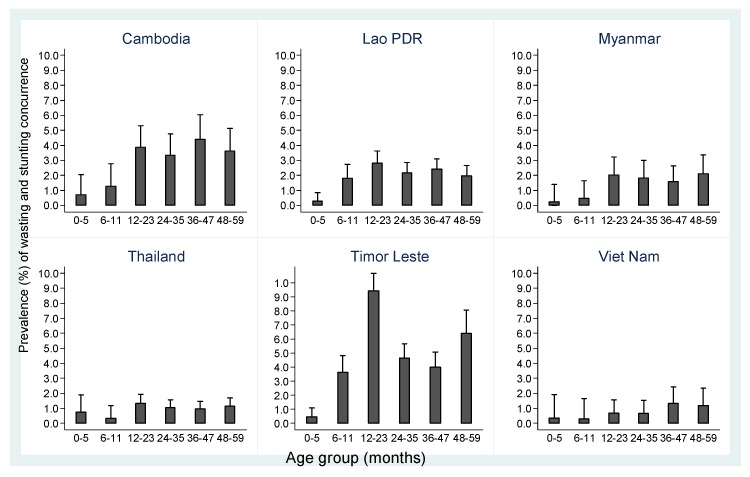
Prevalence concurrence of wasting and stunting by country and age groups.

**Table 1 nutrients-12-00559-t001:** Selected households, mothers and children characteristics by country.

Characteristic	Country						
	Cambodia	Lao PDR ^2^	Myanmar	Thailand	Timor-Leste	Vietnam	Total ^1^
Type of survey	DHS ^3^	MICS ^4^	DHS	MICS	NFNS	MICS	
Year of survey	2014	2017	2015	2015/2016	2013	2011	
Households interviewed, n	15825	22287	12500	28652		11614	
Mothers/caretakers of children under 5 interviewed, n	7165	11812	4815	12250	9443	3729	49214
Total analyzed sample, n	7165	11812	4815	10551	9482	3729	47554
% rural residence in analyzed sample	72.8	69.3	79.0	54.2	74.9	61.4	67.9
% improved water source	67.1	80.7	79.3	97.9	63.8	67.9	80.0
Age of the mother distribution in analyzed sample	*n* = 7165	*n* = 11812	*n* = 4815	*n* = 10351	*n* = 9482	*n* = 3729	*n* = 47554
% <20 years	2.4	6.7	2.2	4.7	3.1	2.3	4.1
% 20–35 years	53.5	72.1	67.7	62.2	65.3	74.4	65.5
% ≥35 years	44.1	21.2	30.1	33.1	31.6	23.3	30.4
Mother’s marital status	*n* = 7165	*n* = 11168	*n* = 4815	*n* = 10453	*n* = 9942	*n* = 3523	*n* = 46566
% in union	94.5	97.5	95.3	93.8	97.2	97.4	95.9
Highest level of education of the mothers in analyzed sample	*n* = 7165	11720	*n* = 4815	*n* = 10500	*n* = 9456	*n* = 3431	
No formal education	14.4	22.9	17.9	0.0	31.0	0.0	
% Primary	49.4	38.9	44.3	3.7	26.1	19.7	
% Secondary	32.4	20.5	31.0	22.0	17.3	60.3	
% Higher than secondary	3.8	17.7	6.8	74.3	25.6	20.0	
Height of the mothers in analyzed sample	*n* = 4703	*n* = 0	*n* = 4757	*n* = 0	*n* = 9271	*n* = 0	
<145 cm	5.9		8.1		13.3		
≥145 cm	94.1		91.9		86.7		
BMI of mothers (kg.m^−2^)	*n* = 4699	*n* = 0	*n* = 4753	*n* = 0	*n* = 9253	*n* = 0	
% <18.5	11.5		10.4		25.7		
% 18.5–25.0	71.3		67.1		64.8		
% 25.0–30.0	14.4		17.8		8.4		
% ≥30.0	2.7		4.7		1.1		
Antenatal care by skilled provider	*n* = 5560	*n* = 11812	*n* = 3311	*n* = 10551	*n* = 7840	*n* = 3729	*n* = 42803
≥4	78.5	58.3	66.1	67.7	76.7	76.4	68.7
<4	21.5	41.7	33.9	32.3	23.3	23.6	31.3
Distribution of sex among surveyed under 5 children	*n* = 7165	*n* = 11812	*n* = 4815	*n* = 10551	*n* = 9409	*n* = 3729	*n* = 47481
% male	50.4	50.9	52.5	51.2	51.1	51.0	51.1
Size at birth	*n* = 7122	*n* = 6170	*n* = 4631	*n* = 4143	*n* = 0	*n* = 1665	
Very large than average	4.1	2.1	1.6	1.8		1.3	
Large than average	31.3	12.6	22.6	17.4		10.2	
Average	53.1	77.2	61.9	70.8		78.7	
Small than average	8.9	7.7	12.5	9.3		8.3	
Very small than average	2.6	0.4	1.5	0.6		1.6	
Age distribution (%) of surveyed under 5 children (in months)	*n* = 7165	*n* = 11720	*n* = 4815	*n* = 10500	*n* = 9460	*n* = 3678	*n* = 47338
0–5	10.0	9.7	10.3	5.9	12.0	8.7	9.3
6–11	10.9	10.4	10.0	6.3	13.4	9.5	10.0
12–23	20.6	18.9	19.6	20.7	25.3	20.7	21.0
24–35	19.8	20.3	19.7	22.3	21.0	21.4	20.9
36–47	18.8	21.4	21.1	23.0	17.1	20.9	20.5
48–59	19.9	19.3	19.3	21.8	11.2	18.8	18.3
Breast feeding practices	*n* = 7153	*n* = 8015	*n* = 4815	*n* = 6265	*n* = 4758	*n* = 3604	*n* = 34610
% ever breastfed	95.4	100.0	97.4	100.0	89.0	100.0	97.2

^1^ total not calculated when data missing for any of the countries; ^2^ PDR = People’s Democratic Republic; ^3^ DHS = Demographic Health Survey; ^4^ MICS = Multiple Indicator Cluster Survey.

**Table 2 nutrients-12-00559-t002:** Pooled and by country prevalence and burden of wasting and severe wasting.

Country	Year of survey	Total under 5 population ^1^	Sample Size	Prevalence (95%CI ^2^)		Burden ^3^ (Number of Affected)	
				Wasted ^4^	Severely Wasted ^5^	Wasted ^4^	Severely Wasted ^5^
Cambodia	2014	1,742,000	4359	10.0 (8.9; 11.2)	2.6 (2.1; 3.4)	174200	45292
Lao PDR^6^	2017	777,000	11362	9.0 (8.3; 9.8)	3.0 (2.6; 3.4)	69930	23310
Myanmar	2015	4,565,000	4200	6.8 (5.9; 7.9)	1.4 (1.0; 1.9)	310420	63910
Thailand	2015/16	3,784,000	9541	5.8 (4.6; 7.3)	1.5 (1.0; 2.4)	219472	56760
Timor-Leste	2013	190,000	9257	10.6 (9.4; 11.9)	2.3 (1.8; 2.9)	20140	4370
Vietnam	2011	7,185,000	3561	4.1 (3.4; 4.9)	1.2 (0.8; 1.7)	294585	86220
Total		18,243,000	42,280	8.9 (8.0; 9.9)	2.0 (1.7; 2.4)	1,088,747	279,862

1 Figures obtained from state of world children reports; ^2^ CI = confidence interval; ^3^ Burden is obtained by multiplying the total under 5 population by the weighted prevalence; ^4^ wasted = weight-for-height<-2 Z-score; ^5^ Severely wasted = weight-for-height <-3 Z-score; ^6^ Lao PDR = Lao People’s Democratic Republic.

**Table 3 nutrients-12-00559-t003:** Prevalence and burden of wasting and stunting concurrence.

Country	Year of Survey	Total Under 5 Population ^1^	Sample Size	Prevalence (95%CI ^2^)		Burden ^3^ (Number of Affected)	
				Wasted & Stunted ^4^	Severely Wasting & Severely Stunted ^5^	Wasted & Stunted ^4^	Severely Wasting & Severely Stunted ^5^
Cambodia	2014	1,742,000	4336	3.27 (2.74; 3.79)	0.14(0.03; 0.25)	56,963	2439
Lao PDR ^6^	2017	777,000	11225	2.22 (1.95;2.49)	0.17 (0.09; 0.24)	17,249	1321
Myanmar	2015	4,565,000	4186	1.56 (1.18; 1.93)	0.05 (0.00; 0.12)	71,214	2283
Thailand	2015/16	3,784,000	9525	1.10 (0.89; 1.31)	0.14 (0.06; 0.21)	41,624	5298
Timor-Leste	2013	190,000	9220	5.30 (4.84; 5.76)	0.31 (0.19; 0.42)	10,070	4370
Vietnam	2011	7,185,000	3552	1.05 (0.71; 1.38)	0.03 (0.00; 0.09)	75,443	2156
Total		18,243,000	42,044	1.65 (1.53; 1.78)	0.13 (0.09; 0.16)	272,563	17867

1 Figures obtained from state of world children reports; ^2^ CI = confidence interval; ^3^ Burden is obtained by multiplying the total under 5 population by the weighted prevalence; ^4^ Wasted & stunted = children with both weight-for-height<-2 Z-score and height-for-age<-2; ^5^ Severe wasted & severely stunted = children with both weight-for-height<-3 Z-score and height-for-age<-3 Z-score; ^6^ Lao PDR = Lao People’s Democratic Republic.

**Table 4 nutrients-12-00559-t004:** Predictors of wasting: univariate and separate multivariate multilevel regression analyses for the six countries.

Risk Factor	*n*	% Wasting	Univariate Analysis				Multivariate Model			
			OR	95% CI		*p*-Value	AOR	95% CI		*p*-Value
				LL	UL			LL	UL	
**Households characteristics model**										
Residence										
Rural	27844	8.03	1.00							
Urban	12969	7.52	0.96	0.87	1.06	0.478	1.06	0.97	1.18	0.172
Number of household members										
<4	4561	8.89	1.00				1.00			
4–6	24417	7.50	0.86	0.77	0.96	0.007	0.84	0.75	0.94	0.003
7–9	10353	8.21	0.84	0.74	0.95	0.005	0.84	0.74	0.96	0.009
≥10	2968	9.01	0.85	0.72	1.00	0.054	0.86	0.72	1.03	0.109
Wealth index										
Poorest	11070	9.38	1.45	1.20	1.76	<0.001	1.11	0.99	1.25	0.063
Poorer	8540	7.85	1.18	0.96	1.45	0.122	0.98	0.87	1.10	0.734
Middle	7545	7.79	1.00	-	-	-	1.00	-	-	-
Richer	7931	6.98	0.88	0.71	1.10	0.275	0.85	0.75	0.96	0.009
Richest	7200	7.06	0.78	0.61	0.98	0.035	0.89	0.78	1.02	0.0
Drinking water source										
Improved	32880	7.55	1.00							
Unimproved	7934	10.91	1.52	1.32	1.75	<0.001	1.36	1.23	1.49	<0.001
**Mothers characteristics model**										
Mother’s Age (years)										
<20	1807	10.25	1.00				1.00			-
20–35	29431	8.05	0.78	0.67	0.92	0.002	0.79	0.67	0.92	0.003
≥35	11070	7.33	0.70	0.59	0.83	<0.001	0.68	0.58	0.81	<0.001
Body Mass Index (kg.m^−2^)										
<18.5	3278	14.97	1.70	1.51	1.91	<0.001				
18.5–25	11765	9.13	1.00							
25–30	2160	6.82	0.75	0.62	0.90	0.002				
≥30.0	419	4.80	0.54	0.34	0.85	0.008				
Mother’s height(cm)										
<145	1781	10.52	1.05	0.89	1.24	0.522				
≥145	15864	9.99	1.00							
Mother level of formal education										
None	6788	10.33	1.00				1.00			
Primary	11974	9.10	0.98	0.88	1.09	0.689	0.96	0.87	1.07	0.482
Secondary	10757	7.11	0.92	0.82	1.04	0.186	0.90	0.80	1.01	0.075
Higher	12507	6.77	0.79	0.70	0.89	<0.001	0.76	0.68	0.86	<0.001
Mother marital status										
Not in union	1598	7.18	1.00							
In union	39994	8.00	1.09	0.89	1.32	0.395	1.10	0.90	1.34	0.341
**Children characteristics model**										
Child sex										
Male	21566	8.55	1.00							
Female	20742	7.29	0.86	0.80	0.92	<0.001	0.87	0.81	0.93	<0.001
Child age group										
0–5 months	3805	10.5	1.44	1.26	1.65	<0.001	1.46	1.27	1.67	<0.001
6–11 months	4297	10.9	1.41	1.23	1.60	<0.001	1.36	1.19	1.56	<0.001
12–23 months	9027	10.37	1.37	1.23	1.53	<0.001	1.33	1.19	1.59	<0.001
24–35 months	8853	6.06	0.87	0.77	0.99	0.031	0.86	0.76	0.97	0.019
36–47 months	8688	5.51	0.83	0.74	0.94	0.004	0.83	0.73	0.93	0.003
48–59 months	7638	6.92	1.00				1.00			
ANC visits by skilled provider										
≥4 by skilled provider	13660	7.45	1.00							
<4 by skilled provider	9175	10.66	1.14	1.02	1.27	0.020				
Other	15805	6.19	0.94	0.85	1.04	0.249				
Size at birth										
Very large	458	4.72	0.53	0.34	0.82	0.005				
Larger than average	3879	6.29	0.67	0.58	0.79	<0.001				
Average	13320	7.40	1.00							
Smaller than average	1823	9.92	1.51	1.28	1.77	<0.001				
Very small	210	10.81	2.06	1.39	3.04	<0.001				
Ever breastfed										
No	1042	10.97	0.95	0.77	1.17	0.652				
Yes	29422	8.36	1.00							
Vaccines up-to-date										
No	5594	8.29	1.00							
Yes	14817	9.70	1.02	0.90	1.16	0.747				
Unsure	9063	5.15	0.68	0.57	0.80	<0.001				
History of fever										
No	32922	7.38								
Yes	9372	9.78	1.26	1.16	1.36	<0.001	1.21	1.11	1.32	<0.001
History of diarrhoea	42232									
No	38111	7.65	1.00							
Yes	4149	10.35	1.27	1.14	1.42	<0.001	1.14	1.02	1.28	0.020
History of Cough										
No	12679	9.53								
Yes	5160	11.00	1.12	1.01	1.26	0.035				

^1^ OR = odds ratio; ^2^ CI = confidence interval; ^3^ AOR = adjusted odds ratio; ^4^ LL = lower limit; ^5^ UL = upper limit.

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
