# Peer review of "The Forgotten Agenda of Wasting in Southeast Asia: Burden, Determinants and Overlap with Stunting: A Review of Nationally Representative Cross-Sectional Demographic and Health Surveys in Six Countries"

_nutrients, 2020, doi:10.3390/nu12020559_

Round 1
Reviewer 1 Report
The paper touches an important topic overall, and argues that wasting in South East Asia is a significant public health concern. A few areas that need to be addressed before the paper can be accepted include the following:
- The authors must include the current neonatal, infant and under-five mortality figures in the paper. Their argument that wasting is a public health risk in these settings (in the Discussion section of the paper) is on the basis of the association between wasting and mortality but they present no data on the mortality in the countries for which they have done their analysis so it is a little hard for the reader to accept that mortality is a significant risk. E.g., Thailand has an IMR of 9 per 1000, which is very small, and most likely primarily neonatal.
If mortality itself is not a big public health risk in the countries the authors choose for the analysis, then can the authors comment on other risks of wasting? e.g., there is a literature on early wasting predicting later stunting, for instance.
- Severe wasting is quite low (1-3 percent) in the countries they are including in their analysis. Can the authors clarify whether facility-based management is not adequate in the context of such low numbers? Many of these countries have fairly well developed health care set ups.
- The authors also are not clear what proven interventions they are recommending for scale-up and whether there is evidence that those might be acceptable or feasible for use in this region. If it is CMAM, see my point above on the low numbers.
Is there any further disaggregated analysis the authors could do about zones within the countries in their regions that have a higher potential prevalence of wasting and severe wasting.In closing, the authors need to be VERY clear about the region- and country-specific current mortality. They should also be careful to not apply mortality risks from other contexts to a regional context which is known for fairly low.
Author Response
Response to Reviewer 1 Comments
Point 1: In the abstract, they concluded wasting is serious problem in the Southeast Asia. As a reader, I request comparison of the prevalence between developed countries and the focused countries
Response 1: Thank you for this comment and suggestion. We have revised the abstract (lines 20 and 21) and the discussion (298) to make the comparison of the prevalence in our study with developed countries.
Point 2: Methods: Definitions of wasting and shunting need to be clearly presented
Response 2: We thank the reviewer for spotting this omission . We have now include a paragraph with the definitions (lines 90-94)
Point 3: Figure 3: The prevalence of concurrence of wasting and stunting varies. Readers would need interpretations of the variation by countries in the Discussion section.
Response 3: Thank you for the recommendation, we have added paragraph has been added to implement this recommendation (line 359-370).
Point 4: Model selection: The authors did not select stepwise method to construct multivariate logistic regression. I partly agree with this selection. However, they need to somehow justify how to select the explanatory variables. I think that there could be limit of available variables. If it is the circumstance, they need to excuse it in the manuscript but strengthen the utilisation of the variables.
Response 4: We thank the reviewer for this comment especially for agreeing that stepwise methods should not be used for the construction of models. We are aware of the existing debate regarding the use or not using stepwise methods for the selection of predictors. We adhere to the approach of not using these methods. We have slightly adjusted the phases to clearly indicate the selection and exclusion procedure.
We have also mention this in the limitations section of the discussion.
Point 5: Discussion: I wonder how wasting is public health problem. General readers need to know wasting and stunting would harm children’s lives in the future and sometimes deprive their lives.
Response 5: We thank the reviewer for this comment however we have not made any change in the text; we believe that this concern is already well addressed in the first paragraph of the introduction.
Point 6: Further, what kind of the other diseases and educational gaps wasting and stunting could cause in childhood and adulthood?
Response 6: We feel that this comment is the same as the previous one and that we have well covered this point in paragraph one of the introduction.
Point 7: In the Discussion and Instruction sections, they could introduce the harm.?
Response 7: We opted not to repeat this point that is already included into introduction section in the discussion to minimize the length of the manuscript.
Point 8: Discussion: The variation of the prevalence among countries could be attributed to economics and culture of common sense or diet?
Response 8: Several studies published in the literature have shown that undernutrition prevalence can remain higher in a country demonstrating tremendous economic growth and this observation is covered in paragraph starting from line 497. We have cited some of these studies (references 36, 44). Given this literature and because all the countries of the region are experiencing rapid economic growth we could not link the difference across countries to economical differences as suggested by the reviewer.
Also, we are not sure about the effect of cultural differences and dietary habit on the country-by-country difference in prevalence. We tend to believe that the most important factors related to food intake will be insufficient intake due to household food insecurity and low dietary diversity score at households’ level. We mentioned these as limitations of our analysis. We believe that the association between some of the covariates included in the models could have changed if we adjusted for food security of the household and dietary diversity score.
Point 9: Readers may not recognise what categories of paper this is (review or cross-sectional study). They could clarify it in the title.
Response 9: We thank the reviewer for the suggestion. We have adjusted the title accordingly.
Reviewer 2 Report
This analysis aimed at determining the burden, risk factors and the level of childhood wasting and stunting among the Southeast Asia region. Data from demographic and health survey and Multiple indicator Cluster Surveys datasets from six countries of the region were analyzed. The pooled weighted prevalence for being wasted and concurrently wasted and stunted among children 0-59 months were 8.9%, and 1.6%, respectively. Predictors for being wasted included source of drinking water, wealth index, urban residence, child’s age and history of illness and mother’s body mass index. The study concluded that wasting is a serious public health problem in the region that should be addressed urgently using preventive and curative approaches. In brief, the descriptive study is well written, and it pointed out the important issues which call for immediate intervention.
Comments:
To make it concise, the texts and paragraphs in the Section of Discussion can be shortened for better understanding of the readers. The style of the references are inconsistent, please make changes for the references to conform to the requirement of the journal.
Author Response
Response to Reviewer 2 Comments
Point 1: Introduction and discussion should be improved by quoting some missing relevant articles. For example, authors should incorporate and discuss this Mapping child growth failure across low- and middle-income countries
Response 1: We deliberately limited the use of evidence and papers from other regions of the world to focus on the main objective of the paper which is to highlight the situation of the Southeast Asia. Thus, the exclusion of papers covering the same topic in other UN regions was intentional and remains our preference.
For the reference specifically mentioned by the reviewer, we confirm that we read it prior to completing our write up as it was published while we were already in the process of submitting the paper. We noted that the with regards to wasting, the region of our focus is not specifically specified in the paper. We also believe that the key messages of the paper are similar to those contained in our manuscript. Thus, we agree that the paper is very important and will be very influential, but we chose not to this paper in our manuscript.
Point 2: Moreover, there are some typos and English mistakes disseminated throughout the text, that should be checked by an English native speaker and carefully proof-read.
Response 2: Thank you, we have made the necessary corrections.
Reviewer 3 Report
The authors investigated prevalence of wasting and shunting. They further analysed risk factors of the outcome. I would like to present comments to improve the manuscript.
I have following concerns.
In Abstract, they concluded wasting is serious problem in the Southeast Asia. As a reader, I request comparison of the prevalence between developed countries and the focused countries. Methods: Definitions of wasting and shunting need to be clearly presented. Figure 3: The prevalence of concurrence of wasting and shunting varies. Readers would need interpretations of the variation by countries in the Discussion section. Model selection: The authors did not select stepwise method to construct multivariate logistic regression. I partly agree with this selection. However, they need to somehow justify how to select the explanatory variables. I think that there could be limit of available variables. If it is the circumstance, they need to excuse it in the manuscript but strengthen the utilisation of the variables. Discussion: I wonder how wasting is public health problem. General readers need to know wasting and shunting would harm children’s lives in the future and sometimes deprive their lives. Further, what kind of the other diseases and educational gaps wasting and shunting could cause in childhood and adulthood? In the Discussion and Instruction sections, they could introduce the harm. Discussion: The variation of the prevalence among countries could be attributed to economics and culture of common sense or diet?
I have the other minor comment.
Readers may not recognise what categories of paper this is (review or cross-sectional study). They could clarify it in the title.
Overall, I think that this manuscript would contribute to public health and medical professionals. I think that its usefulness could be enlarged, if the authors describe the text open to all the readers.
Author Response
Response to Reviewer 3 Comments
Response: We have reviewed all the comments and found them same as those of reviewer one.
Reviewer 4 Report
Thanks a lot for providing me with the unique opportunity of serving as reviewer of this interesting article submitted to the prestigious Nutrients journal for potential consideration and publication.
I have some concerns: introduction and discussion should be improved by quoting some missing relevant articles. For example, authors should incorporate and discuss this Mapping child growth failure across low- and middle-income countries. Local Burden of Disease Child Growth Failure Collaborators. Nature. 2020 Jan;577(7789):231-234.
Moreover, there are some typos and English mistakes disseminated throughout the text, that should be checked by an English native speaker and carefully proof-read.
Author Response
Response to Reviewer 4 Comments
Response: Upon reviewing all review 4 comments, we found them to be the same as those review two. Answers have been provided under reviewer 2.
Round 2
Reviewer 3 Report
I think that the authors have addressed all of my comments. I have no more comment. I appreciate them for their efforts to report the important results.